# The polyketide to fatty acid transition in the evolution of animal lipid metabolism

Zhenjian Lin[1,3], Feng Li[1,3], Patrick J. Krug[2] & Eric W. Schmidt [1] ✉

Animals synthesize simple lipids using a distinct fatty acid synthase (FAS) related to the type I polyketide synthase (PKS) enzymes that produce complex specialized metabolites. The evolutionary origin of the animal FAS and its relationship to the diversity of PKSs remain unclear despite the critical role of lipid synthesis in cellular metabolism. Recently, an animal FAS-like PKS (AFPK) was identified in sacoglossan molluscs. Here, we explore the phylogenetic distribution of AFPKs and other PKS and FAS enzymes across the tree of life. We found AFPKs widely distributed in arthropods and molluscs (>6300 newly described AFPK sequences). The AFPKs form a clade with the animal FAS, providing an evolutionary link bridging the type I PKSs and the animal FAS. We found molluscan AFPK diversification correlated with shell loss, suggesting AFPKs provide a chemical defense. Arthropods have few or no PKSs, but our results indicate AFPKs contributed to their ecological and evolutionary success by facilitating branched hydrocarbon and pheromone biosynthesis. Although animal metabolism is well studied, surprising new metabolic enzyme classes such as AFPKs await discovery.

Fatty acids are required by living organisms, yet strikingly different branches of the tree of life have acquired convergent solutions to fatty acid biosynthesis. The animal fatty acid synthase (FAS) in particular has an independent origin and distinct domain architecture compared to other FASs, even those found in close relatives such as fungi[1]. Instead, the animal FAS has the same domain organization and clear sequence and structural homology with a class of enzymes known as the type I polyketide synthases (PKSs) (Fig. 1A). Widely found throughout the tree of life, including within animals[2–5], type I PKSs produce complex secondary (specialized) metabolites such as antibiotics, pigments, and many other biologically and commercially important compounds[6] (Fig. 1C). While both PKSs and FASs polymerize acetate and its chemical relatives, in contrast to PKSs the animal FAS produces fully saturated lipids (Fig. 1B). The current model is that animal FAS shares a common ancestor with fungal type I PKS[1], but the evolutionary origins of fatty acid biosynthesis in animals remain surprisingly unclear.

Recently, the enzyme EcPKS1 from sacoglossan molluscs was described which seemed to bridge these two types of metabolism.

EcPKS1 is phylogenetically closely related to animal FAS, but instead of saturated fats, it made complex products similar to those produced by PKSs[7]; thus, a potentially new family of enzymes was designated, the animal FAS-like PKSs (AFPKSs). EcPKS1 was part of specialized metabolism, making unique compounds so far found only in sacoglossans, where they are associated with the ability of the animals to perform photosynthesis[8,9]. A provisional phylogenetic analysis indicated that there might be more FAS-like enzymes in molluscs. However, technical difficulties made it difficult to discover new AFPKs, since they are very similar to animal FASs and are often misassembled and/or misannotated in omics databases. In addition, some of these very FAS-like enzymes were identified as FAS paralogs in insects and associated by genetic methods with cuticular (branched-chain) hydrocarbons and pheromones[10–12]. This mystery spurred us to ask whether AFPKs were widespread in the animal world and potential evolutionary intermediates bridging FAS and PKS metabolism. If so, these largely uncharacterized enzymes might explain the vast number of lipid-like molecules found in the animals for which no biosynthetic pathways have been defined.

[1]Department of Medicinal Chemistry, University of Utah, Salt Lake City, UT 84112, USA. [2]Department of Biological Sciences, California State University, Los Angeles, CA 90032, USA. [3]These authors contributed equally: Zhenjian Lin, Feng Li. ✉e-mail: ews1@utah.edu

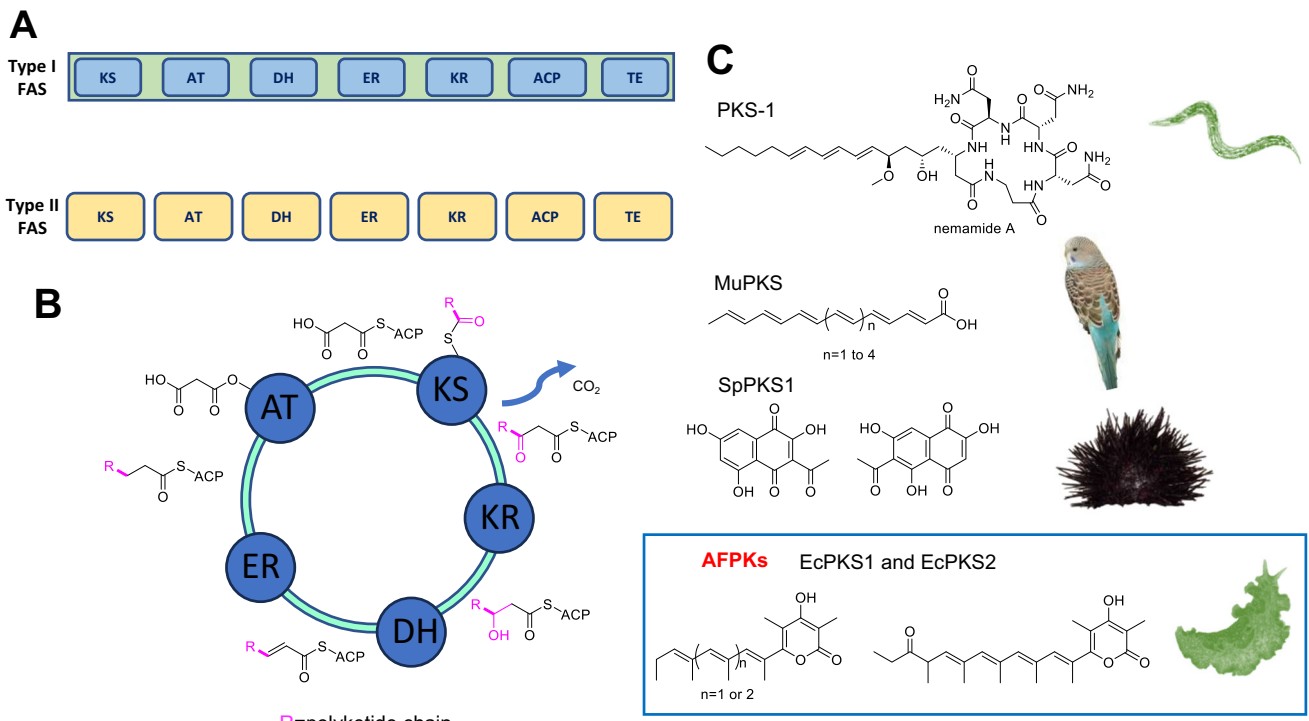

**Fig. 1 | Lipid and polyketide biosynthesis in the animals (metazoans). A** The canonical functional domain architecture of the animal type I fatty acid synthase (FAS) is shown, with one of its major products, the fully saturated lipid oleic acid. Below, the type II FAS, such as those found in mitochondria and in bacteria, plants, and elsewhere, are encoded on individual proteins. **B** The catalytic cycle of a FAS or polyketide synthase (PKS) enzyme. The acyltransferase (AT) loads substrates, most commonly malonate, onto the acyl carrier protein (ACP). Substrates are then condensed by the ketosynthase (KS), creating an elongated product with a ketone. The ketoreductase (KR) reduces the ketone to an alcohol, while the dehydratase

(DH) eliminates water to produce the olefin. Finally, the enoylreductase (ER) reduces the olefin to a fully saturated lipid. The R group on the growing lipid is methyl in the starter unit and becomes elongated in further iterative reaction cycles. **C** Known animal PKS and animal FAS-like PKS (AFPK) products. Note that variable substrates (methylmalonate) can be used and that variable reduction can lead to ketones, polyenes, and other features not normally synthesized by the animal FAS. The animal PKSs and AFPKs reported to date have similar domain architectures to the animal FAS, with exceptions at their C-termini. An AFPK product is shown in the box, while PKS products are outside the box.

Here, we developed bioinformatics methods that reliably differentiate mitochondrial type II FASs, animal type I FASs, PKSs, and the phylogenetically intermediate AFPK enzymes. We demonstrate that AFPKs are widespread in molluscs and in arthropods but rare or absent from other animal taxa investigated. AFPKs share a common ancestor with the animal type I FAS, indicating a single origin early in the development of the animal phyla. Further, these results reinforce previous ideas that fungal type I PKSs and animal FAS share a common ancestor. Finally, the methods clarified the phylogeny of KS-containing enzymes in the animals, revealing key aspects of their evolution, origin, and distribution. In some cases, patterns clearly reflect biological and ecological roles, while for the most part, these are new and uncharacterized enzymes. Taken together, these results reveal an unexpected enzymatic repertoire across major animal phyla that may underlie much of the chemical richness of diverse groups. The designation of AFPKs as a distinct group is supported by their presence in a derived clade with animal FAS on the global KS tree, coupled with the distinctive biochemical features of the AFPKs characterized to date, which together distinguish the AFPKs from canonical animal PKSs.

## Results

### Obtaining PKS and FAS sequences from across the animal kingdom

We aimed to globally identify AFPKs in animals, but we anticipated four technical problems. First, animal FAS/PKS proteins are often poorly assembled due to their lengths and, sometimes, the presence of multiple closely related copies in a genome/transcriptome[5]. Therefore, our

workflow involved downloading all sequence read archives (SRAs) from GenBank for taxa of interest (Supplementary Data 10 and Supplementary Table 1), (re)assembling them and then performing further analyses. Because of our interest in their elaborate polyketide chemistry[13], we also sequenced a representative of *Siphonaria* (NCBI accession numbers: SRR22547485 and SRR22547486) and added its transcriptome and genome to the same workflow as used for SRAs[14]. Second, many animal datasets contain sequences originating in co-occurring organisms. Contaminating contigs from bacteria, fungi, plants, and algae were removed by the taxonomy assignment pipeline in the Autometa package[15]. Third, type I FAS/PKS are multidomain enzymes (~500 kDa in the dimeric state) that are difficult to align, except for the N-terminal ketosynthase (KS) domains. Thus, we analyzed only KS domains. Finally, with EcPKS1 and EcPKS2 as the only biochemically characterized AFPKs to the best of our knowledge[7,16], it was difficult to identify and distinguish AFPKs from FAS enzymes. To solve this problem, we first developed profile hidden Markov models (HMMs) using animal PKSs and FASs that were previously identified[5,7]. To mitigate any potential bias in HMM scores for KSs originating from various animal species, the training of HMMs incorporated FAS sequences from a diverse range of species, spanning the phyla Cnidaria, Nematoda, Annelida, Tardigrada, Mollusca, Echinodermata, and subphylum Vertebrata. Using these profiles, we employed the distribution pattern of HMM scores to automate the identification of animal cytoplasmic and mitochondrial FAS, PKSs, and related enzymes in animal datasets. Simultaneously, we distinguished between hypothetical AFPKs and animal type I FAS. Subsequent phylogenetic analyses supported the identification of AFPKs. We used only full-length,

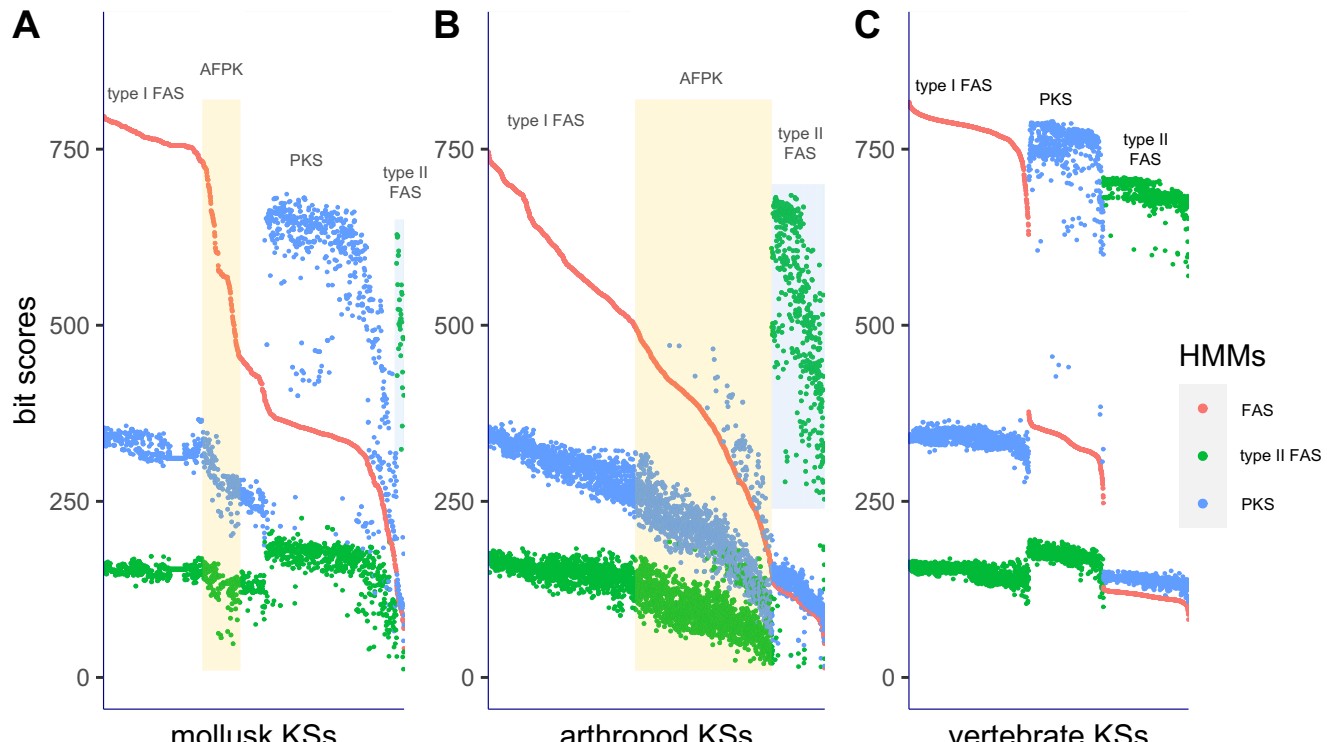

**Fig. 2 | Alignment bit scores of the ketosynthase (KS) domains to profile hidden Markov models (HMMs) (animal fatty acid synthase (FAS), polyketide synthase (PKS), and mitochondrial FASII).** In this method, we noticed a discrete region bridging the animal type I FAS and the animal PKSs. The yellow bar shows a region in which putative animal FAS-like PKSs (AFPKs) dominate; for the hmm scores in the order FAS, PKS, FASII for EcPKS1 (587.7, 274.6, 123.1) and EcPKS2 (581.7, 285.2, 133.5); these were further evaluated and validated using additional methods including phylogenetic analyses. **A** Mollusk KSs; **B** Arthropod KSs; **C** Vertebrate KSs. The x-axes indicate sequence number, ordered by relationship to cytoplasmic FAS; the y-axes indicate HMM bit scores.

well-assembled KSs with a validated origin in animal genomes for all further analyses described in this study.

## Widespread distribution of diverse, KS-containing type I enzymes in animals

Profile HMM analysis using 558 mollusc transcriptome assemblies led to the identification of contigs potentially encoding KS-containing enzymes. Nonredundant KS-containing genes were then used as queries to search against the nr database in NCBI to identify further mollusc KS-containing genes. These were initially employed to identify 1390 KS-encoding enzymes in Mollusca, as well as 18,039 KS-encoding genes in >5000 specimens from Arthropoda (representing the two major protostome lineages). The algorithm was used to unveil all KS-containing enzymes in 896 sponge transcriptome assemblies from the SRA database (phylum Porifera). It was also applied to Chordata (482 transcriptomes from 282 species from subphylum Vertebrata, as well as 232 transcriptomes from subphylum Tunicata) for comparison. Sponges were chosen to represent basal animals given sufficient genomic and transcriptomic resources for our analyses compared to other candidate groups (e.g., Ctenophora); vertebrates were used to represent deuterostomes given their relevance to human physiology and medicine. Finally, we identified FAS genes in other representative metazoan taxa.

Strikingly, while mitochondrial type II FAS enzymes were readily identified in Porifera, we could not find other KS-containing proteins encoded within sponge genomes. Our pipeline is designed to differentiate sponge-encoded genes from those in the abundant bacteria that are often present, making this a robust analysis. Previously, type I FAS was found to be very rare or absent in sponges[17] and type I FAS genes identified in our analysis appear to originate in dinoflagellate symbionts. Our result uses a much larger dataset but is otherwise consistent with these previous analyses. We also failed to detect type I FAS in the limited available ctenophore datasets. In contrast, all other animal groups investigated, from placozoans to chordates, harbor the cytoplasmic FAS. Further, animal FAS could not be identified in the choanoflagellates, thought to be the sister group of animals[18]. This implies that the animal FAS may have originated in the ParaHoxazoa[19].

Although type I FAS-like enzymes were identified in molluscs[7,16], it was initially difficult to differentiate true FAS enzymes from the AFPKs, especially because robust phylogenetic tree methods are not practical at the scale needed to parse omics data. Therefore, we applied an HMM method. Three different HMMs (animal type I FAS, PKS, and mitochondrial type II FAS) were generated using protein sequences from GenBank (Supplementary Data 1–3). The alignment scores of the KS domains were compared in each of the three models. The data was visualized in a dot plot in comparison to the ordered FAS HMM alignment bit scores (Fig. 2). We focused on the discrete region bridging the animal type I FAS and the animal PKSs in the HMM score dot plot, with scores for EcPKS1 and ECPKS2 helping to roughly reference the region. Although not as accurate as phylogenetic analysis, this method was extremely rapid and enabled us to readily differentiate potential AFPKs, PKSs, and type I and II animal FASs. These methods were applied to the available data from molluscs, arthropods, and vertebrates.

In the molluscs, using EcPKS1 and EcPKS2 as references, the type I FASs, type II FASs, AFPKs, and PKSs could be readily distinguished visually, with abundant AFPKs discovered across molluscan groups. The identified AFPKs were used in further analyses that confirmed their distinctness from related FASs (see below).

Similar to molluscs, the arthropods appeared to contain a large number of extremely diverse, unanticipated AFPKs. However, unlike what was observed in molluscs, there was not a clear distinction

between FASs and AFPKs; instead, a continuum was observed spanning the FAS–PKS transition. This made it difficult to use this method to firmly define AFPKs (further details below), as could be done with molluscs. Further, very few PKSs were identified in arthropods.

The chordate KS-containing proteins revealed very clear patterns reflecting PKS and lipid diversity in the group. No AFPKs were discovered in any available vertebrate transcriptome, genome, or the GenBank nr database, despite exhaustive searches using this HMM method, maximum likelihood (ML) phylogenetic analyses, and even manually screening datasets. In contrast, 2014 FAS or PKS genes were detected in vertebrates. Thus, AFPKs are not universal in animals but may be restricted to protostomes. We also investigated all tunicate SRAs (another group of chordates), finding that they contained only animal FAS, and not PKSs or AFPKs. By contrast, PKSs are relatively widespread among vertebrates, including many found in mammals. Only one of these vertebrate PKSs has been characterized: a bird PKS that synthesizes polyenes, coloring budgerigars green[20]. The roles of other vertebrate PKSs are unknown, except that a fish PKS with an unknown product is required for otolith (ear) formation[21]. No PKSs were seen in any placental mammal; all were in marsupial genomes, implying that PKSs might have been lost in the transition to eutherian mammals. Overall, the vertebrate data expands previous knowledge of PKSs and reveals many proteins of unknown function or biological significance.

PKSs are broadly distributed in the animal kingdom, present in every phylum investigated except for sponges and ctenophores (Supplementary Fig. 1). Aside from vertebrate proteins mentioned above, PKSs have been characterized in phylum Echinodermata[5], where at least one makes aromatic pigments, and in phylum Nematoda, where a *Caenorhabditis elegans* PKS-nonribosomal peptide synthetase makes a complex hormone/signaling compound[2]. In contrast to the PKSs, AFPKs were only identified in molluscs and arthropods.

To further demonstrate that the HMM model applied to vertebrates and that we were not missing something due to the model sequence set, we examined how different KS training sequences from less diverse species affected the medium score of AFPKs. To do this, we downloaded all vertebrate protein sequences annotated as FAS or PKS

from GenBank. We extracted the KS domains, removed redundant sequences, and subjected them to analysis using an ML tree. We only detected FAS and PKS genes in our analysis. Subsequently, we constructed new HMMs for FAS and animal PKS using the newly detected sequences exclusively from vertebrates sourced from GenBank. These HMMs were then utilized to generate HMM score dot plots for molluscs and vertebrate KSs. Remarkably, the resulting HMM score plots are very similar to the ones in Fig. 2 (Supplementary Fig. 5), reinforcing the robustness of the analytical method. For example, only the FAS HMM scores for EcPKS1 and EcPKS2 are ~80 lower in comparison to the corresponding values in Fig. 2, but these two KSs are still in a discrete region bridging the animal type I FAS and the animal PKSs.

## PKSs and AFPKs are prevalent in molluscs and form six major clades

We focused on Mollusca because AFPKs have been biochemically characterized[7,16] from sacoglossan molluscs, a group of sea slugs including chloroplast-retaining species[8,9] in which AFPK products are likely to be important for photosynthesis. To better define the phylogenetic diversity of molluscan AFPKs, two methods were performed. First, we picked the region shown in Fig. 2 with FAS HMM scores from 400 to 600, based on the scores of EcPKS1 and EcPKS2; potential AFPK protein sequences were selected and then aligned with randomly selected molluscan FAS and PKS sequences. The resulting alignment was used to create an ML tree. Phylogenetic analysis differentiated molluscan FASs, AFPKs, and PKSs into seven different clades (Fig. 3A). Two of these clades represented the canonical animal type I FAS and PKS groups. Outside of those groups were four clades (mo-clades 1-4) that were more similar to FAS than to PKS, which we categorized as AFPKs. The functionally characterized EcPKS1 and EcPKS2 reside in mo-clade 1. mo-clade 1 proteins are closely related to the animal FAS, even though they make products that are much different than expected from FAS chemistry. The mo-clade 1 AFPKs produce partially reduced pyrone polyenes derived from methylmalonate[7,16] instead of linear, saturated fats derived from acetate that are produced by FASs. It was not initially clear whether mo-clade 5 was more FAS-like or PKS-like, but it appeared to be more closely related to the animal PKSs than were the other AFPK clades.

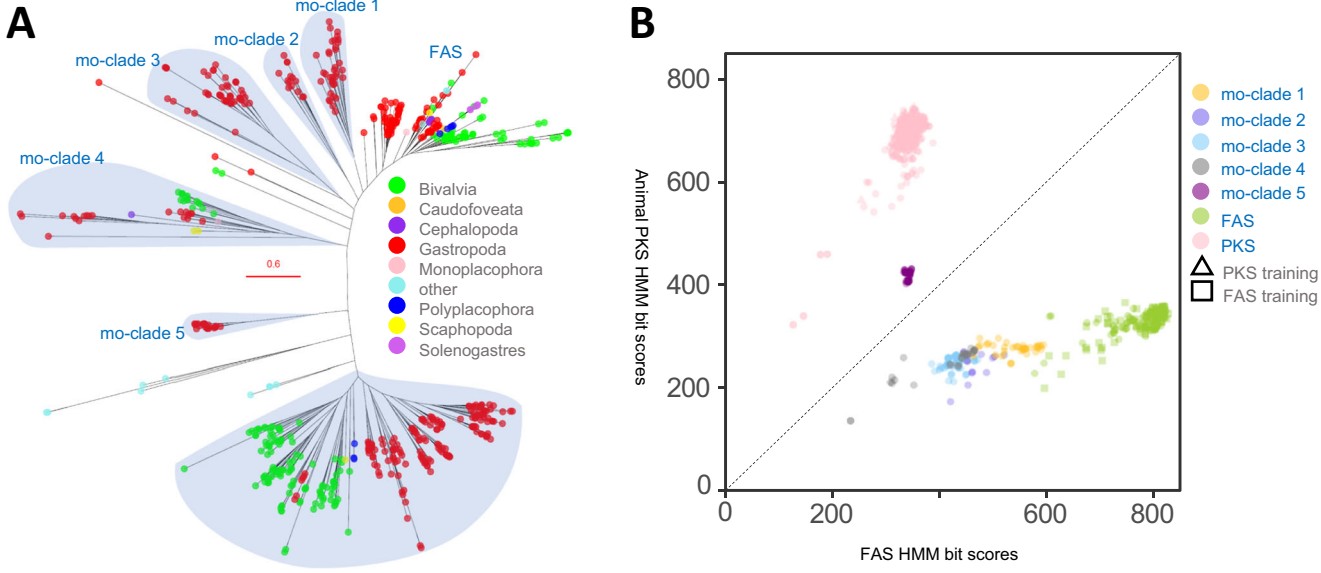

**Fig. 3 | Analysis of ketosynthase- (KS)-containing proteins from molluscs. A** Maximum-likelihood phylogenetic trees of fatty acid synthases (FASs), polyketide synthases (PKSs), and animal FAS-like PKSs (AFPKs) from molluscs (Supplementary Data 4). **B** Hidden Markov model (HMM) alignment bit scores of the ketosynthase (KS) domains reveal different subclasses of mollusc PKS, AFPK, and FAS (Supplementary Data 5 and 6).

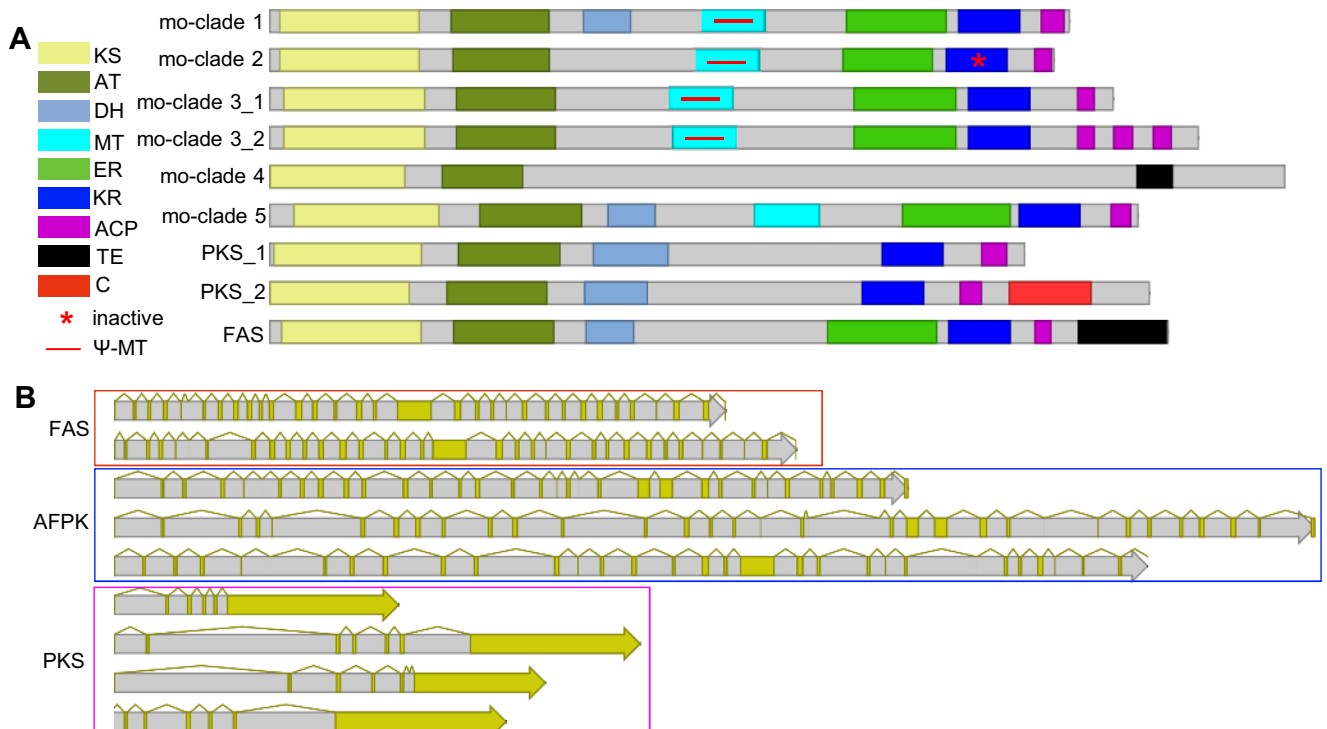

**Fig. 4 | Gene and protein organization in the molluscs.** Domain architecture (**A**) and intron pattern (**B**) of fatty acid synthases (FASs), polyketide synthases (PKSs), and animal FAS-like PKSs (AFPKs) found in molluscs. KS ketosynthase, AT acyltransferase, DH dehydratase, MT methyltransferase, ER enoylreductase, KR ketoreductase, ACP acyl carrier protein, TE thioesterase, C condensation.

With the goal of creating a practical, accurate, and rapid method for categorizing AFPKs, we randomly selected FAS and PKS sequences from the initial phylogeny (Fig. 3A) to create two training sets: one from the FASs, and one from the PKSs. The training sets were used to generate HMMs, which were applied to analyze all mollusc KSs (Fig. 3B). The HMM score of each KS-containing protein sequence was plotted in a scatter graph, where any protein sequence above the line $y = x$ is more closely related to PKSs, while AFPKs and FASs are below the line. Using these models, we identified 113 nonredundant putative AFPKs in mo-clades 1–4 from existing mollusc SRA datasets. Because mo-clade 5 was above the line comprising $y = x$, it was tentatively identified as a PKS clade.

The domain architecture of KS-containing proteins was predicted by antiSMASH[22] and Interpro[23] (Fig. 4). All animal FAS proteins contain a thioesterase (TE) domain that is responsible for hydrolyzing the final product. However, AFPKs in mo-clades 1–3 lacked a TE domain. In the case of mo-clade 1 proteins EcPKS1 and EcPKS2, offloading is accomplished without a TE, possibly by the spontaneous formation of a pyrone ring system[7]. However, for the majority of these proteins, the offloading mechanism is unknown.

Strikingly, no domain was predicted by antiSMASH for the protein sequences in mo-clade 4, while Interpro was only able to predict the KS-AT-TE domains. The majority of mo-clade 4 enzymes lack predictable sequence similarity with other proteins, indicating as yet unknown biochemistry. All but one of the clades had ketoreductase (KR) domains predicted to be active by the algorithm in antiSMASH[22]. Animal FAS enzymes contain pseudo-methyltransferase (Ψ-MT) domains that are structurally important but catalytically inactive[24]. They may have evolved from the active MT present in fungal type I PKSs. Underscoring the close relationships between FAS and AFPKs, many AFPKs retain identifiable Ψ-MT domains. mo-clade 5 is the only one that is likely to contain active MT domains, further supporting its phylogenetic placement amongst the PKSs and not the AFPKs. mo-clade 2 proteins were predicted to encode inactive KR domains; such

proteins should likely lead to the formation of aromatic compounds. Tandem acyl carrier protein (ACP) domains were detected in some of the proteins in mo-clade 3. In summary, the AFPKs have diverse domain architectures, the chemical products of which are currently unpredictable.

The mollusc PKS clade contains two distinct domain architectures: those with condensation (C) domains from nonribosomal peptide biosynthesis, and those without C domains (Fig. 4A, Supplementary Data 11). In taxa (i.e. *Siphonaria*) that contain both types of PKSs, the KS portions are phylogenetically very closely related. Overall, this result and the domain architecture differences seen in AFPKs suggested that the N-terminal regions encoding the KSs are relatively conserved, while the C-terminal regions arise through recombination at least in some cases.

From the available genome sequences, we observed that the exon density of mollusc PKS genes is much higher than that found in FAS and AFPK genes. The few introns observed in mollusc PKS genes were concentrated at the N-terminus (Fig. 4B, Supplementary Data 12). In contrast, AFPK and FAS genes have high intron densities through their entire lengths, with the exception of rather large exons in the Ψ-MT domains. Based upon their position between conserved and variable parts of the proteins, these Ψ-MT domain regions might be sites of recombination.

## Origin of AFPKs (mo-clades 1–3) is highly correlated with shell reduction in gastropod molluscs

In the 1081 KSs detected from 558 mollusc SRA assemblies, there are 525 FASs, 178 AFPKs and 378 PKSs (including canonical PKSs + mo-clade 5). Up to nine KSs were found in a given species (Fig. 5A), but the distribution of AFPK/PKS clades differed drastically among SRA samples. Plotting clade distribution by molluscan class and genus (Fig. 5B), mo-clades 1–3 were only detected in a few genera within Gastropoda, while other mo-clades are widely distributed. Strikingly, mo-clades 1–3 do not co-occur with PKSs in most analyzed genera.

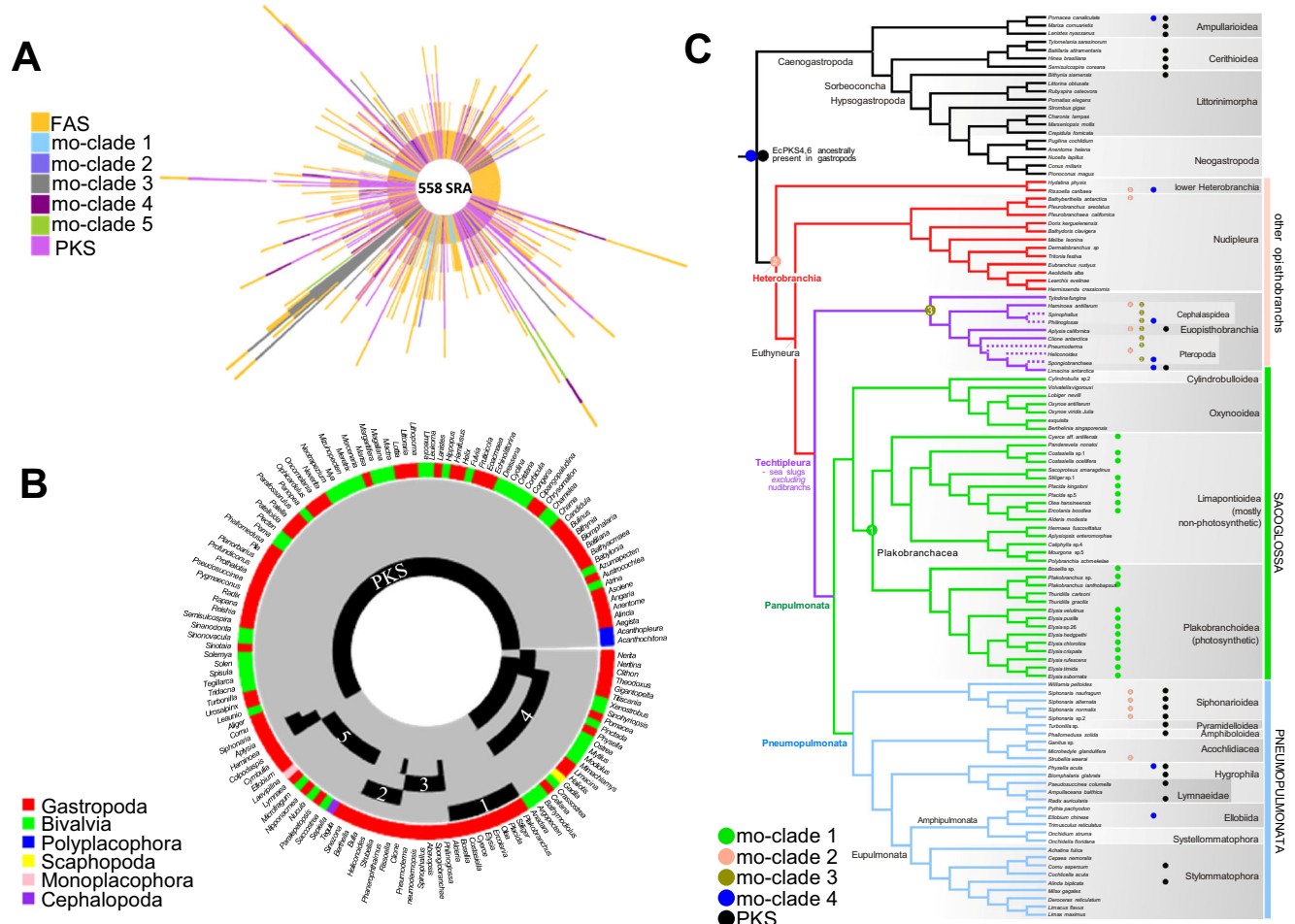

**Fig. 5 | Distribution of polyketide synthase (PKS) and animal fatty acid synthase- (FAS)-like PKS (AFPK) genes in molluscs.** **A** Bar plot of the counts of ketosynthase-(KS)-containing genes detected in each of the 558 SRA specimens. The bar color indicates the KS clade. **B** Heat map plot of AFPK and PKS clades distribution in 143 genera analyzed. The inner black ring indicates the clade detected in each genus. The outside ring color indicates the class to which the genera belong. **C** Mapping AFPKs and PKSs to a current gastropod phylogeny. The colored circles indicate each AFPK/PKS clade. A colored circle on top of a node in the phylogenetic tree indicates that the AFPK/PKS clade is found only within that taxonomic group.

Phylogenetic evidence indicated that the repertoire of AFPK diversity increased in concert with progressive reduction of the ancestral shell in Heterobranchia, a major gastropod lineage including sea slugs and traditional pulmonate (air-breathing) snails and slugs (Fig. 5C). Indeed, almost all molluscan families known to contain polypropionate or polyene polyketides contained AFPKs (mo-clades 1–3) (Supplementary Fig. 2)[25]. Canonical animal PKS enzymes were sampled from most major molluscan lineages (Polyplacophora, Gastropoda, Bivalvia, Scaphopoda), while mo-clade 4 AFPKs were present in all surveyed molluscan classes including Cephalapoda and Monoplacophora (Fig. 5B, C). mo-clade 4 AFPKs were also expressed in all major gastropod subclasses (Patellogastropoda, Vetigastropoda, Neritimorpha, Caenogastropoda, and Heterobranchia). In contrast, mo-clades 1–3 were phylogenetically restricted to Heterobranchia (Fig. 5C). Major evolutionary trends within Heterobranchia were (a) convergent reduction and loss of the ancestral shell (a physical defense) in many 'sea slug' lineages, and (b) the transition to air-breathing and invasion of freshwater and terrestrial habitats in Pneumopulmonata, culminating in the explosive radiation of stylommatophoran snails and slugs. By facilitating the biosynthesis of small molecules used as anti-predator defenses, sunscreens, and in other chemical signaling roles, AFPKs may have facilitated shell loss and the colonization of novel habitats in heterobranchs, which comprise about one-third of molluscan species diversity.

Within Heterobranchia, mo-clade 4 enzymes were sampled in lower heterobranchs, euopisthobranch sea slugs, freshwater snails (Hygrophila), and amphibious members of Amphipulmonata, sister group to the terrestrial Stylommatophora[14] (Fig. 5C). mo-clade 2 was found only in Heterobranchia, but was present in diverse groups: the lower heterobranchs; a pleurobranch; euopisthobranchs including the model organism *Aplysia*; and basal pneumopulmonates, including *Siphonaria* and an acochlidiacean. This distribution indicates the ancestor of mo-clade 2 was present in the most recent common ancestor of Heterobranchia. In contrast, mo-clade 3 AFPKs were only sampled in Euopisthobranchia: cephalaspideans (bubble shells and kin), sea hares (e.g., *Aplysia*), and pteropods (sea butterflies). Euopisthobranchia had the richest repertoire of biosynthetic potential, expressing enzymes from three mo-clades as well as the canonical animal PKS lineage. As euopisthobranchs underwent repeated, parallel reductions in the ancestral shell and radiated into habitats including the pelagic realm[26], further exploration of the potential role of polyketides in defense and adaptation to planktonic life is warranted[27–30]. The second phylogenetically restricted AFPK lineage was mo-clade 1 AFPKs, expressed solely in shell-less sacoglossans (clade Plakobranchacea). These sea slugs expressed only mo-clade 1 AFPKs, and many of the species analyzed had multiple AFPKs within this lineage.

Alternative chemical defenses may have selected against AFPK expression or gene retention. Strikingly, no PKS or AFPK genes were

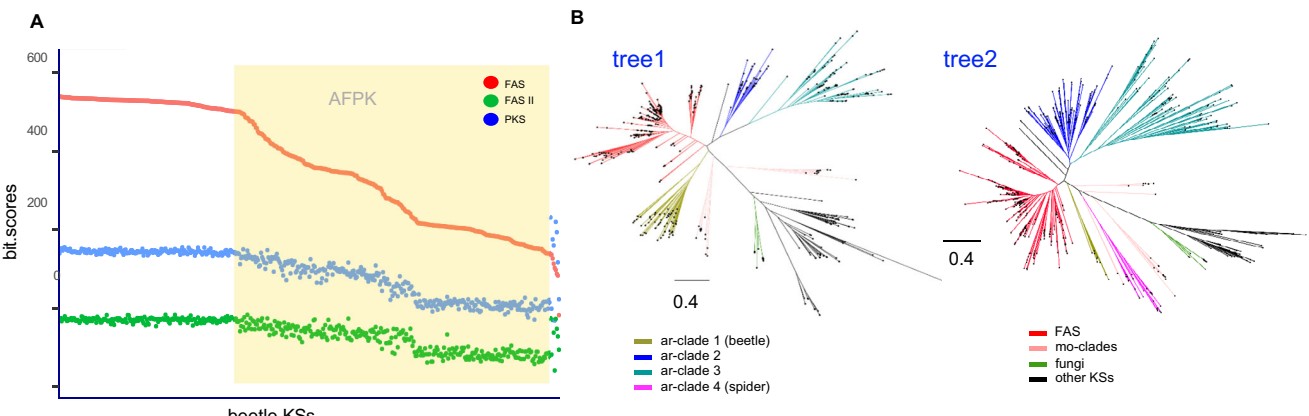

**Fig. 6 | Distribution of animal fatty acid synthase-(FAS)-like polyketide synthase (AFPK) lineages among arthropods based on the phylogeny and identity of available ketosynthases (KSs). A** The hidden Markov model (HMM) score distribution of beetle KSs has a similar trend to that for the whole arthropod KSs shown in Fig. 2B but with a clearer breakpoint from the FASs. **B** Congruence between beetle KS tree (tree1) and selected arthropod KS tree (tree2). The "other KSs" include PKSs from bacteria and molluscs, ar-clades1 and 4 are close to mollusca AFPKs. Alignment files are provided in Supplementary Data 7 and 8.

detected in nudibranchs, which typically deploy diet-derived chemicals or cnidarian nematocysts for defense, in lieu of a shell[27,31]. mo-clade 1 AFPKs were not detected in the shelled sacoglossans (superfamily Oxynooidea). The shells in this group are thin and likely provide little defense, but most species store defensive compounds from their host, the "killer algae" *Caulerpa*[32,33]. The only other major group lacking PKS expression was the Neogastropoda, in which complex venoms and a heavy shell may have favored the loss of ancestral polyketide chemistry[34]. These phylogenetic trends further implicate a role for AFPKs and the compounds they produce in defensive strategies, and highlight the interplay between phenotypic tradeoffs, genome evolution, and diversification dynamics across Gastropoda.

## AFPKs from arthropods

Arthropods contained numerous potential AFPKs but they were more difficult to distinguish from FASs than were the mollusc AFPKs, necessitating refinement of methods (Fig. 2B). The KSs originated in 2622 different arthropod species, and as a result, the observed sequence diversity was much greater than found in mollusc and vertebrate data sets. For this reason, the slope of the FAS HMM bit score was virtually continuous, without discrete transitions between enzyme classes as in the mollusc and vertebrate analyses. We hypothesized that the arthropod KSs with FAS HMM bit scores between 500 and 200 (Fig. 2B, shaded region) comprised AFPKs. However, this area included some PKS genes (Fig. 2B, red dots above the green line). Compared to the majority of AFPKs in the area, those PKS genes had higher PKS HMM scores. For example, one of them was previously identified as a horizontally acquired PKS gene[35] (GenBank accession: OXA62418.1) in the springtail *Folsomia candida* genome, which has HMM scores in the order FAS, PKS, FASII: 272.0, 325.6, 132.6 in the plot. We hypothesized that, as found in *Folsomia*, many arthropod PKS genes in this region of the plot potentially result from horizontal gene transfer. We therefore predicted much of the polyketide repertoire of arthropods likely derives from the biosynthetic activity of AFPKs, which subsequent analyses revealed are widespread in arthropods.

To resolve the arthropod AFPKs, we first took a subset of the data, comprising all 477 KSs from beetle species. The dot plot of HMM bit scores (Fig. 6A) showed a very similar trend to that for the whole arthropod KSs, but with a sharper break point between the FAS and AFPK sequences. It also revealed at least two different types of AFPKs in beetles, since there are two regions with different slopes. Indeed, evolutionary relationships among 477 beetle KSs (Fig. 6B, tree1, Supplementary Data 7) supported three KS clades (ar-clades 1–3) that are

phylogenetically distant from the FAS clade common among animals. The HMM scores of the KSs from these three clades suggested that they are AFPKs (Figs. 2B and 6A).

To determine whether this pattern was recapitulated throughout arthropods, the KSs (>5000) from SRA assemblies were sorted according to the FAS HMM score, and every tenth sequence was selected to provide 497 KS sequences that were analyzed by ML (tree 2 in Fig. 6B, Supplementary Data 8). The resulting phylogeny is highly congruent with the beetle KS tree (tree1 in Fig. 6B), with two of the AFPK clades (ar-clades 2 and 3) distributed throughout the arthropods. ar-clade 1 was only sampled in beetles, and therefore had reduced representation on the all-arthropod tree (tree2 in Fig. 6B). The ar-clade 1 lineage may be restricted to the spectacular radiation of beetles, one of the major sources of terrestrial biodiversity[36–38], and thus warrants special attention given the unknown role of these enzymes[39,40]. In addition, a fourth AFPK clade (ar-clade 4) was identified only in spiders, another exceptional animal radiation[41–44]. This spider clade was closely related to mollusc AFPKs, reflecting the ancient origin of AFPKs prior to the divergence of major bilaterian lineages. Because many arthropod AFPKs are very closely related to FASs, we took a subset of sequences from each of the AFPK clades (ar-clades1–4) identified in tree2. These subsets were used as reference points to better understand the distribution patterns of all arthropod KSs. The dataset consisted of a total of 6542 nonredundant KS sequences obtained from selected arthropod SRA datasets and the GenBank nr database. These sequences were randomly divided into 11 groups, each containing approximately 600 KS sequences. Combining each of these groups with the corresponding reference sequence, we conducted a thorough analysis using ML methods (see Supplementary Fig. 3). The resulting phylogenetic trees showed remarkable consistency across the set of 11 trees and when compared to tree1 and tree2 presented in Fig. 6b. Notably, the reference sequences for ar-clades1–4 were distributed in all of the major clades of the phylogenetic trees; thus, these four ar-clades represented all major AFPK lineages detected in available arthropod transcriptomes.

Arthropod AFPKs have a similar domain architecture (including a TE domain) to that found in FASs. A few of the sequences we investigated had alternative termination domains, including the reductive (R) domains that often terminate fungal and bacterial PKS and peptide synthetase enzymes[45]. In many cases, this implies a much more complex lipid metabolism in these animals than is currently appreciated; potentially many of the unusual lipids isolated from arthropods might originate from the activity of as-yet uncharacterized AFPK sequences[46]. These include ethers, aldehydes, alcohols, and branched-chain lipids[47].

We found that some of those in ar-clades 2 and 3 have been previously associated with insect-specialized metabolism. For example, in *Locusta migratoria*, there are three different type I FAS orthologs that are annotated as "FAS"[12]. Knockout and expression studies showed that one LmFAS2 (QNU13193), which we recovered in the FAS clade, is expressed systemically as the normal type I FAS, while the other two, LmFAS1 (QNU13192, ar-clade2) and LmFAS3 (QNU13194, ar-clade3) were expressed in the integument. Knockout of LmFAS1/LmFAS3 altered the cuticular hydrocarbon and/or inner hydrocarbon profile. Paralogous "FAS" enzymes were similarly associated with specialized metabolism in several other insects[10,11], but based on our findings those genes are predicted to be AFPKs. We hypothesize that the biochemical characterization of arthropod AFPKs will reveal the source of many unusual lipids and hormones distributed throughout the phylum.

## Model of FAS and AFPK common origin and evolution in animals

Applying the HMMs used above in this study, we identified only mitochondrial (type II) FASs in sponges and ctenophores. In neither group did we find the type I FAS/AFPK/PKS enzymes. By contrast, we identified type I FASs in all phyla of ParaHoxozoa. Both ctenophores and sponges are noted for the prevalence of fatty acid elongases[48], which makes the unique suite of lipids known only in the sponges. In higher animals and yeast, the mitochondrial FAS is specialized to produce octanoic acid needed for lipoate biosynthesis[49]. We hypothesize that sponges and ctenophores might use the type II FAS to produce short-chain octanoate, which is matured by cytoplasmic elongases[50]. Alternatively, an unknown lipid biosynthetic route may yet be found in these animals or their microbial symbionts[17]. By contrast, the ancestor of ParaHoxozoa used a specialized type I FAS, not found in any other lineage or domain on the Tree of life, to synthesize long-chain lipids.

To further investigate the evolutionary history of AFPK diversity, we inferred the relationships among AFPKs, FASs, and PKSs from a range of eukaryotes (animals, fungi, amoebae) as well as archaea and eubacteria (Fig. 7, Supplementary Data 9, Supplementary Fig. 4 and Supplementary Data 13). The resulting phylogeny reinforces previous suggestions that the animal FAS shares a common ancestor with fungal highly-reducing PKSs)[1]. However, animal FAS shared a more recent common ancestor with the AFPKs, which formed a grade paraphyletic with respect to animal FASs (Fig. 7). The animal FASs were a derived clade nested within the AFPKs, most closely related to ar-clades 2 and 3. These findings suggest an ancestral fungal-like type I PKS was retained in animals and diversified into the AFPK/FAS enzyme family. Based on their phylogenetic distributions, AFPKs and animal FAS likely diverged in the ancestor of ParaHoxozoa. Apparent paraphyly of AFPKs with respect to FAS could be an artifact of rooting, or it may reflect the diversification of AFPKs in speciose radiations promoted by the ecological roles of polyketides while constraints of primary metabolism limited FAS diversity. However, our findings demonstrate that AFPK and FAS lineages share a much more recent common ancestor than either share with PKS enzymes, and suggest a shared evolutionary history of enzyme function between primary and secondary metabolism during animal evolution.

The ML phylogeny also supported a sister relationship for mollusc mo-clade 5 and ameba PKSs, reenforcing that clade 5 belongs to the PKS cluster, and not to AFPKs. We propose that the "mollusc" clade 5 might actually originate in a symbiotic organism living in the host molluscs; alternatively, it could be a true molluscan PKS.

## Fast and efficient identification of AFPKs

Through the use of HMM score sorting methods followed by extensive phylogenetic tree analysis, we identified hundreds of AFPKs, forming eight different clades (mo-clades 1–4 and ar-clades 1–4). Nonetheless, this process was highly time-consuming and heavily dependent on the

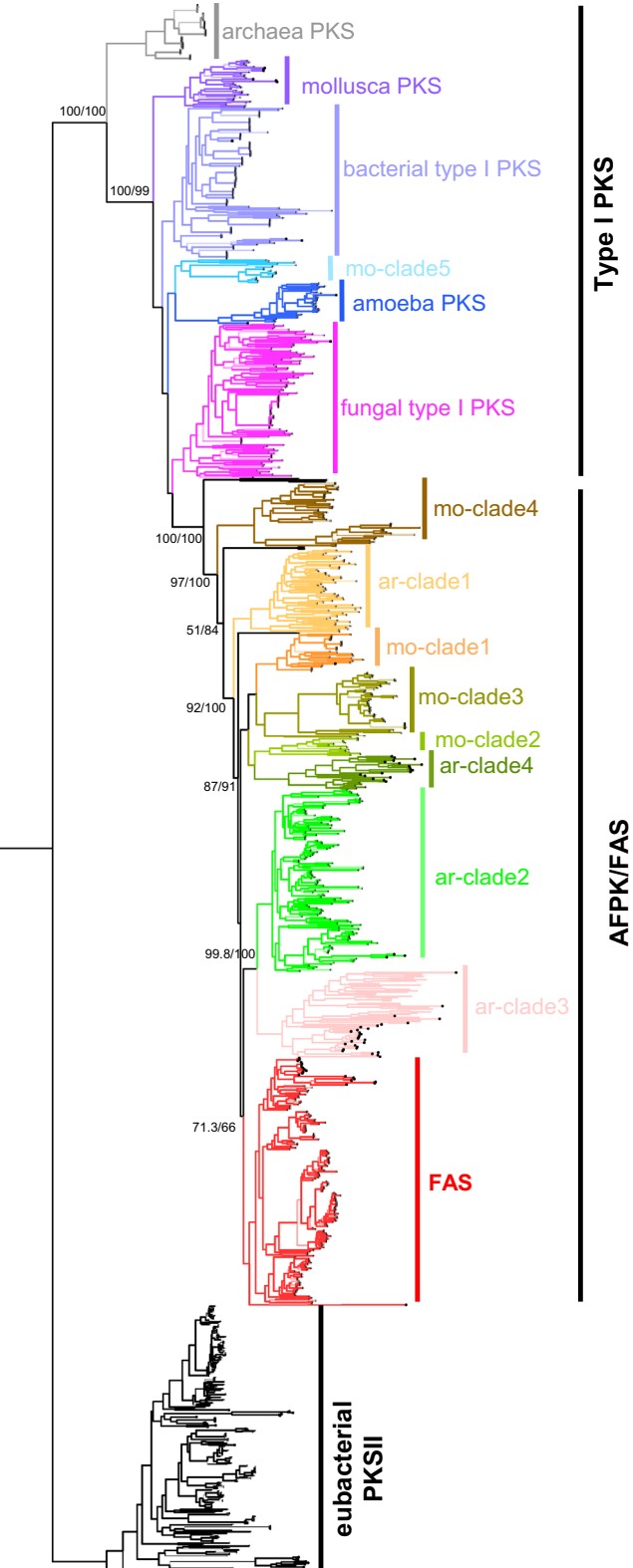

**Fig. 7 | Maximum likelihood tree of selected ketosynthase sequences revealed that the animal fatty acid synthase (FAS) might derive from an animal FAS-like polyketide synthase.** Nodes were supported by the Shimodaira-Hasegawa like-lihood ratio test and ultrafast bootstrap, given as percent values. Alignment files are provided in Supplementary Data 9.

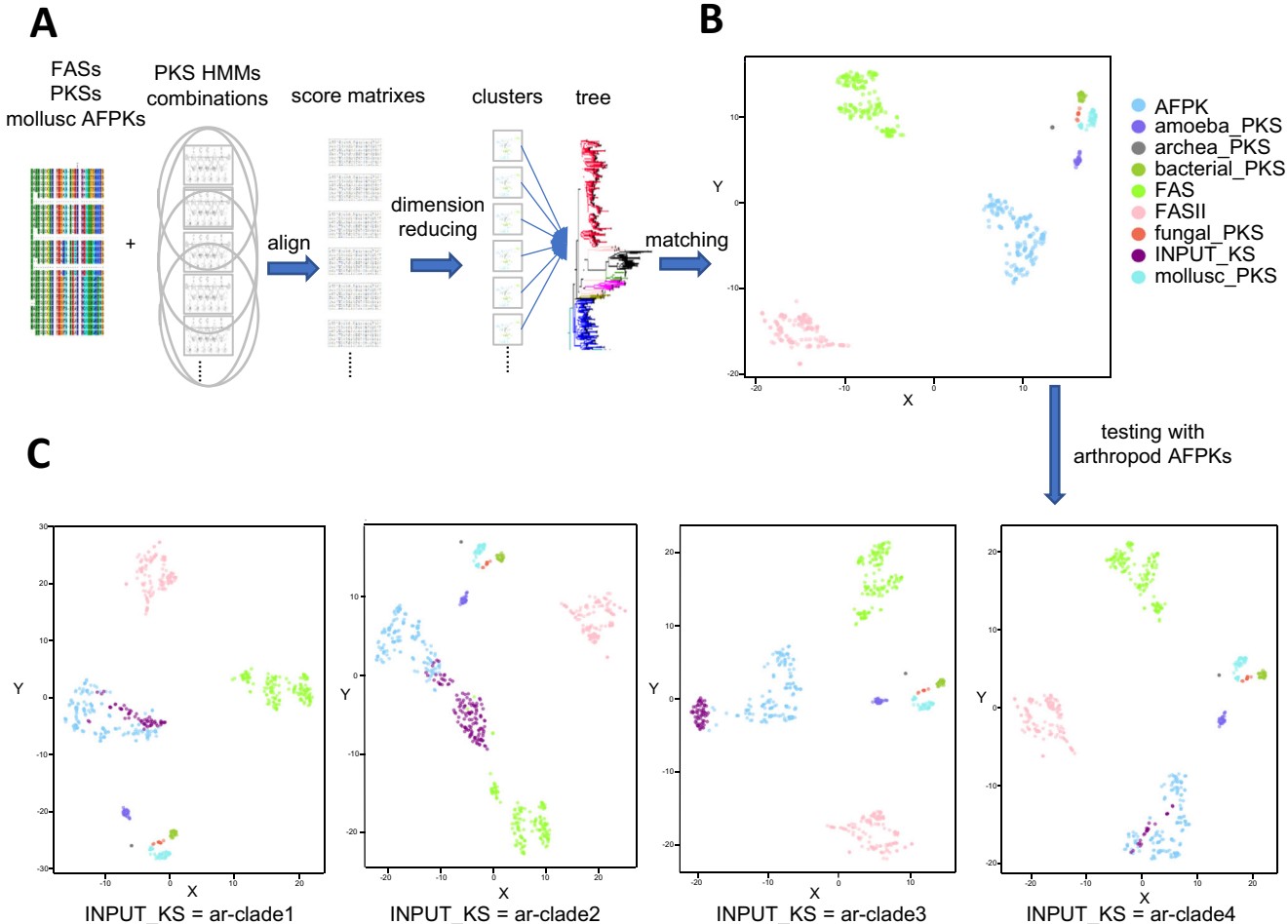

**Fig. 8 | Overview of animal FAS-like polyketide synthase (AFPK)-Finder. A** The training process involved using mollusc AFPKs to align with different combinations of polyketide synthase-related hidden Markov models, and the resulting data matrices of alignment scores were analyzed for dimension reduction. The resulting clusters were evaluated by comparison to the clades in the phylogeny tree in Fig. 3A. **B** The correct model was selected based on the congruence between the clusters and clades. **C** The model was then tested using arthropod AFPKs (purple INPUT_KS) that were identified in Fig. 6.

precise alignment of hundreds of protein sequences, which often required manual curation. We aimed to develop a model using well-defined AFPKs described above to rapidly ascertain the probability that a given KS domain sequence is an AFPK. Such a method would be widely useful in delineating the unexpectedly rich and complex lipid and polyketide metabolism found in animals.

Considering the above limitations, we created AFPK-Finder (DOI:10.5281/zenodo.10125497) to rapidly distinguish AFPKs from PKS and FAS sequences with excellent computational efficiency. First, we prepared a panel of different HMMs using two different resources: we downloaded PKS-related sequences from different organisms from NCBI, and PKS-related HMMs/conserved domains from Pfam and CDD. Next, we used the AFPKs (mo-clades 1-4) identified from mollusks as training data and aligned them to a random subset of the HMMs. The resulting data matrix, which contained the HMM alignment scores, was then normalized and analyzed using Rtsne for dimension reduction (Fig. 8A). The Rtsne 2D plot generated from 30 HMMs showed that the datapoints clustered exactly according to their PKS types (Fig. 8B). This finding indicates that although a KS may not show significant alignment with an HMM, it still provides useful information that helps to annotate its function. We evaluated the model's robustness by utilizing the arthropod AFPKs as the test dataset. ar-clades 1–4 were submitted to AFPK-Finder. The sequences from ar-clades 1, 3, and 4 clustered very well with mollusc AFPKs. Most of the sequences in ar-clade 2 formed their own cluster, with only a small subset of them clustering well with

mollusc AFPKs. In comparison with other ar-clades, ar-clade 2 cluster is much closer to the FAS cluster, which is consistent with the observation in trees 1 and 2 in Fig. 6B. This suggests that ar-clade 2 might have a function very similar to FAS. Furthermore, ar-clade 2 pulled four mol-lusc FAS sequences out of the FAS cluster in the plot, indicating that mollusks may also contain the same type of proteins (Fig. 8C). It is possible that the small number of these sequences makes it difficult to observe a separate clade in the phylogenetic tree.

Overall, AFPK-Finder precisely and rapidly recapitulated our findings from the much more time-consuming HMM-phylogeny ana-lysis shown in Figs. 2 and 3 above. No AFPKs defined in the more rigorous method were missed by AFPK-Finder, but the algorithm rapidly identified a few new AFPKs that were not observed in our initial survey. Moreover, AFPK-Finder readily distinguished and classified all KSs found in animal transcriptomes. Thus, it should be broadly useful in understanding lipid metabolism in the animal kingdom.

## Limitations

There are several limitations to this study. First, in several cases, we have interpreted an absence of a gene or pathway from taxonomic groups in our analysis as indicating a true absence of those genes. This could also result from several other causes, such as a limited sample set, a lack of expression in the tissues analyzed, or the presence of unanticipated orthologs that are not accounted for in the models. Nonetheless, due to the large number of samples, KS types, and

sequences in this study, the overall trends identified are likely to be robust. A second limitation is that our current state of knowledge is not complete, and lipid/polyketide biogenesis and evolution are exceptionally complex. For example, some mussels (molluscs) contain two different varieties of type I FAS. The two FASs share a common ancestor in the phylogenetic tree, but they are not very similar (<50% identity) in protein sequence in comparison to the FAS isoforms detected in other species. Potentially, one of these could be a specialized enzyme arising from within the FAS clade. If validated in further work, such evolution downstream of the AFPK/FAS branch point would indicate a more complex evolutionary pathway to diverse lipids than reflected in the current study, which reflects available sequences and our present understanding of PKS/FAS biochemistry.

## Discussion

Here, we show that polyketide biosynthesis in arthropods and molluscs is likely dominated by AFPKs, a family of proteins that spans the phylogenetic gap between the type I PKSs and the animal FASs. AFPKs and animal FASs form a single clade, with AFPK subfamilies diversifying in specific molluscan and arthropod lineages. Overall, from available transcriptome data, the sum of the methods described above led to the identification of 6122 AFPKs in arthropods and 277 in molluscs. In the few cases where their functions are known, AFPKs in sacoglossan molluscs and in insects contribute to specialized metabolism, producing unusual polyketide-like lipids that are ecologically important to the producing animal. Their biochemical features are intermediate between those of the animal FAS and the PKSs. For these reasons, we propose that the AFPKs comprise a single, true family of KS-containing enzymes.

While polyketide metabolites are well studied in bacteria, fungi, and plants, in animals they represent a largely overlooked group with significant future potential. The methods presented here will enable the biochemical interrogation of this widespread enzyme class and its role in the biology, ecology, and diversification of animals, especially given the association between AFPK diversity and species richness in several major radiations.

## Methods

### Ethics statement

This research complies with ethical regulations. No institutional approval was required for this research.

### RNA extraction and transcriptome sequencing

Live specimens of *Siphonaria* sp. were purchased and shipped from AlgaeBarn.com to the University of Utah in aquarium bags with seawater, inflated with oxygen. The shell was removed, and the whole animal was cut into small pieces (<2 mm²) and homogenized in sterilized nuclease-free water. RNA was extracted using TRIzol (Invitrogen) followed by a DNA-free DNA removal kit (Invitrogen). The quality of the extracted total RNA was evaluated by electrophoresis and QC RIN using the Agilent RNA screen tape assay. An Illumina library was prepared at the Huntsman Cancer Institute's High-Throughput Genomics (HCI-HTG) facility at the University of Utah using Illumina TruSeq Stranded mRNA Library Preparation Kit with polyA selection, and sequenced using an Illumina NovaSeq 6000 sequencer with a ~450 bp insert size and 150 × 150 bp paired-end runs to produce 100M read-pairs. Raw reads were trimmed and adaptors removed by trimmomatic[51], then assembled using rnaSPAdes[52]. Genes were predicted using Prodigal[53] in metagenome mode.

### Genome sequencing

*Siphonaria* gDNA from the homogenized tissue was extracted using the Qiagen DNeasy Blood & Tissue Kit. Illumina library preparation and sequencing were performed at the HCI-HTG. Sequencing library preparation was performed using an NEBNext Ultra II DNA Library Prep Kit with a 450 bp mean insert size. Sequencing used an Illumina NovaSeq

6000 sequencer with 2 × 150 bp runs. Raw reads were trimmed and adaptors were removed by trimmomatic and then assembled using metaSPADES[52]. The animal genes were predicted using AUGUSTUS 3.3[54] with the transcriptome assembly as training data.

### SRA data preparation

SRA fastq raw reads were downloaded from NCBI and assembled using rnaSPAdes. All available gastropod SRA datasets available as of January 2022 were downloaded. For non-gastropod molluscs, the SRA data was sorted in the SRA Run Selector by the Bytes column. Only the top two SRA in byte size were selected for each species and then downloaded. For arthropods and vertebrates, only one SRA data set (the top one in the bytes size) for each species was downloaded. Raw reads for each SRA data were trimmed and adaptors removed by trimmomatic, then assembled using rnaSPAdes. The genes in each assembly were predicted using Prodigal. SRA datasets with low quality were removed if they did not contain at least one KS-containing protein. SRAs used in this study are listed in Supplementary Information.

### Phylogenetic analysis

Orthologous genes were aligned using t-Coffee[55] (-mode mcoffee -output = msf, fasta_aln). To remove poorly aligned regions, the resulting alignment was subsequently trimmed with Clipkit[56] with model parameter '-m kpi-gappy'. The trimmed alignment was then manually inspected to remove any remaining poorly aligned regions. The maximum-likelihood tree was constructed using iqtree[57] (./iqtree -nt AUTO -st AA -alrt 1000 -bb 1000). The ML tree was visualized using ggtree library[58].

### Profile HMM building

To generate KS profile HMMs for type I FASs and animal PKSs, seed sequences were selected from previously identified sequences and from well-annotated animal sequences from GenBank. KS domains of these sequences were predicted using antiSMASH. These KS sequences were used as a query to blastp search against the SRA protein data prepared above. Top hits from the blastp search were analyzed using an ML tree with the seed sequences. The KSs that clade with the seed sequences (FAS and PKS) were, respectively, aligned using t-Coffee (-mode mcoffee -output = msf, fasta_aln) to make HMMs using 'hmmbuild' in the hmmer3 package[59]. Other HMMs were generated using the standard method for 'hmmbuild'.

### KS-containing protein identification

The SRA protein database was searched with the KS HMMs (FAS and PKS) prepared above. A bit score = 180 was set as the threshold for a KS hit. To remove any contamination from the SRA transcriptome assemblies, the corresponding contigs that contain the KS hits were analyzed using the taxonomy assignment pipeline in the Autometa[15] package (make_taxonomy_table.py -a ks_hit_contigs.fa -l 700). The output '.lca' file gave taxonomy ID for the lowest common ancestor of each contig. Based on the taxonomy ID, contigs for bacteria, fungi, plants, and algae were removed. The KS domains of the remaining contigs were predicted by antiSMASH and InterPro. To access KS-containing proteins from GenBank, manually selected, full-length KS domains from the SRA KS hits in each phylum were used as query to search against a standalone nr database, with an output format (-outfmt '6 qseqid sseqid pident length qcov qlen slen mismatch gapopen evalue bitscore staxids sscinames scomnames sskingdoms sblastnames stitle sseq'). The nr hits were filtered using a threshold 'qcov>85' and the staxids matching the animal phylum demand. Here, the nr hit sequences was extracted directly from the 'sseq' output. For each nr sequence ID, there are multiple hits in the blastp output; only the one with the longest 'sseq' was chosen. Finally, the KSs from both the SRA database and no. database were combined, and the duplicated

sequences were removed by Sequence Dereplicator and Database Curator (SDDC)[60].

### KS-HMM bit score analysis

KS domain protein sequences were aligned to different HMM models using hmmsearch function in the HMMER3 package (http://hmmer.org/), with the output option 'hmmsearch --tblout'. The full sequence score for each KS sequence in the output was used for further comparison in dotplot/scatter plot using gglpot library[61].

### Rtsne analysis

Parameters for data normalization and perplexity selection were based on the Rtsne.r script in YAMB[62].

### Reporting summary

Further information on research design is available in the Nature Portfolio Reporting Summary linked to this article.

## Data availability

The alignment files for HMMs and trees are provided in the Supplementary Information. KSs sequences from molluscs, arthropods, and vertebrates used in this study are deposited in figshare: https://doi.org/10.6084/m9.figshare.24066234. Raw sequencing data for the mollusc *Siphonaria* is available in genbank (SRR22547485 and SRR22547486). The original data for plotting in figures is provided in Source Data file. The lists of SRA accession numbers that were used in this paper are provided in Supplementary Data 10. Source data are provided with this paper.

## Code availability

The code for AFPK-finder is available on GitHub (https://doi.org/10.5281/zenodo.10125497).

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

## Acknowledgements

We thank the Center for High Performance Computing, University of Utah, for computational support and J.P. Torres for critically reading the manuscript. This work was funded by NSF IOS 2127111 and 2127110.

## Author contributions

E.W.S., P.J.K., and Z.L. designed the research; Z.L. and F.L. designed the strategies for KS sequencing data collection and analysis; Z.L. per-formed the experiments and analyzed the data. Z.L. and F.L. developed the AFPK-Finder tool; P.J.K. performed the study of the current gastro-pod phylogeny; E.W.S., Z.L., and P.J.K. wrote the paper.

## Competing interests

The authors declare no competing interests.
