## [Peer Review File · Nature Communications]

The polyketide to fatty acid transition in the evolution of animal lipid metabolismREVIEWER COMMENTS

Reviewer #1 (Remarks to the Author):

This manuscript seeks to identify fatty acid synthase-like polyketide synthases (AFPK) and understand their relationship to Type I Fatty acid synthase and Type I polyketide synthases in animals. The structural similarity between animal fatty acid synthase and Type I polyketide synthases is striking, and it is a pleasure to see an analysis of their evolutionary relationship.

Overall, the data and approach in this manuscript are compelling, but the description of the background and significance of the conclusions is too limited.

In particular, the title suggests that the paper will focus on the PKS origin of animal fatty acid synthase. This idea is supported by Figure 7, which includes the mollusk AFPK. However, the tree in Fig. 7 does not include the arthropod AFPK or type II fatty acid synthase. Rooting the tree based on PKS from Archaea and bacteria seems sensible, but it assumes, rather than tests, the idea that animal FAS is derived from an ancestral PKS. If there is data in the literature supporting the PKS origin of animal FAS, then it should be described more fully in the introduction. If not, the authors should add additional analysis to reject the null hypothesis that enzymes involved in specialized metabolism (PKS) evolved from enzymes in central metabolism. Alternatively, the authors could change their title to reflect the main focus of the manuscript, the identification of the AFPKs.

In addition, the manuscript either lacks sufficient background discussion supporting the identification of AFPKs as a separate class of polyketide synthases, or they underemphasize the significance of identifying this group of enzymes. It is unclear from the introduction whether they are merely expanding the identification of a previously known class of PKS enzymes or demonstrating that AFPKs, of which only one or two examples are known, are a widespread, uncharacterized subclass of PKS enzymes. The latter appears to be the case, given that the only AFPK discussed in the introduction was discovered by the authors' lab. Furthermore, it is unclear whether the AFPKs should be considered together as a class separate from the Type I PKS. In the phylogenies and dot plots (like fig. 6A), the AFPKs form multiple clades rather than clustering together. There was also little discussion about the relationship between the arthropod and mollusk AFPKs. Aside from the unrooted trees in Fig. 6, little was done to determine if these two groups of AFPKs have a common origin or evolved independently in different animal lineages.

Addressing the weaknesses listed above and those in the list below will provide clarity to this otherwise very interesting work.

Below are specific questions and comments that should be addressed:

Major comments:

p. 2, abstract, line 6: I don't understand the sentence "Vertebrates lack AFPKs, but excepting the placental mammals contain PKSs." Overall, the abstract also doesn't clearly differentiate between what was previously known and what this manuscript aims to demonstrate, as mentioned regarding the introduction above.

p. 2, introduction paragraph 2: This paragraph suggests that AFPKs are a known class of enzymes that are distinct from Type I PKS, but the reference appears to only identify one or two from a single species. It also suggests that the AFPKs are a single group of enzymes that are distinct from the PKS enzymes, but their domain structure and data in Fig. 2 and 7 raise questions about whether they should be considered a subtype of the Type I PKS. In addition, the introduction suggests that the AFPKs are a single class of enzymes, but the data don't clearly show whether the mollusk and arthropod AFPKs have a single origin, since the arthropod clades are not in Fig. 7.

p. 4 and figure 2: The description of Figure 2 implies that the AFPKs are a known class of enzymes that are clearly defined by the data shown in Figure 2, suggesting that the purpose of the figure is to test an approach to annotate AFPKs, rather than experiment to see if AFPKs are likely to be abundant. When describing Figure 2, wouldn't it be more accurate to say that, in contrast to vertebrates, a significant number of KS domains in arthropods and mollusks score poorly against the 3 HMMs, suggesting that AFPKs are widespread in these species? Because you acknowledge that this method has lower power to discriminate among clades than phylogenetics, shouldn't the results be described as putative or probable AFPKs?

Figure 2 clearly shows that a lot of KS domains have intermediate scores between FAS and PKS HMMs, but is the data really sufficient to define cutoffs to declare which ones are AFPKs, as opposed to more divergent FAS or PKS? In arthropods, some domains within the AFPK group have better PKS HMM scores than FAS scores, and some domains labeled as PKS in the mollusk group also have low scores for the PKS HMM.

The rationale for defining the score boundaries between AFPKs and FAS or PKS should be stated, particularly for arthropods.

Also, could the lower HMM scores of putative AFPKs in mollusks and arthropods be due to technical rather than biological reasons? For example, using a larger number or a less diverse set of vertebrates in the HMM training sets could weight the HMMs to more accurately identify vertebrate FAS and PKS and result in intermediate scores in the other organisms. Lower scores against the HMMs could also arise from truncated KS domains arising from incomplete transcriptome data. How was this taken into account? Could the observation that domains with intermediate scores are absent from the vertebrate data be due to the higher quality of that data?

p. 4, paragraph 3, regarding the statement that no AFPKs were identified in vertebrates: It would be better to say that no KS domains had intermediate bit scores to FAS I, PKS or FAS II, supporting the absence of AFPKs in these species. (Alternatively, the HMMs were weighted toward vertebrate proteins, so the scores are higher)

p. 7, paragraph 4, in the sentence "This spider clade was closely related to mollusc AFPKs, reflecting the ancient origin of AFPKs prior to the divergence of major bilaterian lineages": I am not convinced that the relationships among AFPK clades are clear among these lineages in figure 6b. It does not appear that the spider clade is monophyletic with any particular mollusc AFPK clade or that the evolutionary distance between them is shorter than between other clades (the branches leading to both groups are long).

p. 7, paragraph 4 and figure 6C, around the sentence "Although there are some overlapping regions between clades, the FAS KSs mainly localized in the right top of the plot.": The discussion about fig. 6C is problematic. Ar-clade 3 obviously is more separated (divergent) from FAS because the y-axis HMM was constructed from ar-clade 3 sequences. What is most noticeable about figure 6C is that there is poor separation between FAS and ar-clades 1, 2 and 4, indicating that they are not more similar to ar-clade-3 than FAS. This fits with the phylogeny in fig 6b, where the clades of AFPKs do not cluster together. Aside from clade 3, the other clades and FAS are nearly or completely on the diagonal, indicating that all of them (including FAS) score nearly equally well with both HMMs. Other than showing that the subdivision of the arthropod FAS and AFPKs are ill-defined (also as shown in fig 2b), it is unclear how to interpret this figure or what conclusions should be drawn from it. In addition, the authors should comment on the grey dots, most of which score poorly against either HMM. Are these more divergent groups of FAS or AFPKs? Are they PKS?

Minor comments:

Supplemental data files: It would be helpful if the supplemental data sets were given informative names. Accession numbers of the sequences in the data sets should be included.

Fig. 2: mark where EcPKS 1 and 2 are in fig 2a.

p. 4 paragraph 3: Fig.2c is labeled as vertebrate KS, not chordate. If that is correct, then it is unclear why non-vertebrate chordates (tunicates) are discussed in this paragraph.

p. 4 paragraph 3, second to last sentence concerning eukaryotic parasites. The data supporting the statement that PKS in eutherian mammal data sets are actually from parasites needs to be shown and described, particularly since the accession numbers that are shown list the source as sperm whale.

p. 5 paragraph 2: It is not clear what the "remaining sequences" refers to. Are you referring to sequences whose FAS HMM scores were <400 and >600? Or are you referring to sequences with scores between 400-600 that were not used to build the phylogeny? What were the upper and lower bounds?

p. 5 last paragraph through the beginning of page 6: The discussion of methyltransferase domains is confusing. An MT domain is only noted in clade 5. Are you saying that partial/degraded MT domains are detectable in other clades? If so, then these pseudogene exons should be labeled in fig 4b in order to evaluate and support the last few sentences of this paragraph.

p. 7, paragraph 2, in the sentence about horizontal gene transfer: What is the justification for claiming that some putative AFPKs in arthropods are derived from horizontal gene transfer?

p. 7, paragraph 3, in the sentence "The dot plot of HMM bit scores (Fig. 6A) showed a very similar trend to that for the whole arthropod KSs, but with a much slower descent.": Rather than "slower descent", the wording in the figure legend is more clear - there is a sharper breakpoint between FAS and likely AFPK.

p. 15 figure legend 1: change "are encoded on individual domains" to "are encoded on individual proteins"

p. 15 figure 1B: Some of the chemical structures don't convey the chemical reaction being described. In particular, the substrate of KS is shown as malony-ACP, (3 carbons), and its products are shown as CO₂ + a 4-carbon beta-ketoacyl-ACP (5 carbons), without explaining where the other carbons come from. Also, the product of ER is implied to be a polyunsaturated fatty-acyl-ACP, instead of a fully reduced fatty-acyl-ACP

p. 20, figure 6 legend. In the sentence "Scatter plot of the arthropod KS bit scores to arthropod AFPK ar-clade 3 KS HMM and FAS HMM, showing that selected KSs are representative of the diversity of KSs found throughout the arthropods." Given the large number of low-scoring sequences in gray, which were not included in the phylogeny, the selected KSs used in the phylogeny don't seem that representative of the arthropod KS domains.

p. 21, figure 7: the left section of the tree is labeled AFLP instead of AFPK.

Reviewer #2 (Remarks to the Author):

This manuscript provides new information on the distribution of polyketide synthase and fatty acid synthase genes across broad scope of life forms. In particular, it focuses on animal fatty acid-like polyketide synthases or AFPKs. These biosynthetic systems generate a diverse spectrum of secondary metabolites thought to be important for self-protection, signaling and other communication functions.

The work will be of high significance to the field of natural product sciences, data science/computational analysis of metabolic systems, biochemists, and chemists. The work is highly original and provocative. It may very well provide a road more for many future studies to investigate pathway gene expression, enzymology of the transformation, and further predictive tools to understand the chemical diversity programmed by these fascinating PKSs.

There is some speculation relating to evolution of these systems, but the work provides important hypotheses and conceptual starting points for future studies that demand further detailed studies.

No fatal flaws were identified. While, the noted correlation between loss of mollusk shell and evolution of protective secondary metabolites seems logical, tying it directly to specific types of gene loss beyond the shell would provide a deeper, more sophisticated analysis. Overall the methodology is sound and meets the standards in the field. Sufficient detail is provided in the methods section.

The Conclusion paragraph is excessively brief. It requires more scholarly discussion about the major findings and limitations of this study.

We thank the reviewers for their detailed analysis of the manuscript. Below, we have positively addressed all concerns. We performed the experiments suggested by reviewer 1, all of which supported the initial conclusions. In this revision, we define a surprisingly vast number (6,300+) of previously unnoticed AFPKs, and provide strong support for a model in which the newly defined AFPK and FAS enzyme families form a clade sister to the fungal type I PKSs. We appreciate the suggestions to perform the additional work which made our story that much clearer. The major changes are summarized briefly, along with the rationale:

- 1) The abstract, introduction, and conclusion sections were modified in response to suggestions about contextualizing the results to highlight the novel findings described here.
- 2) We added a section at the front of “Results” (“Obtaining PKS and FAS...”) that better describes our overall pipeline. This was in response to several concerns, especially about a question of whether we used full-length KSs (we did); lines 82-97.
- 3) We better describe and interpret Figure 2 in the text and legends. This was a method used in data exploration that enabled us to generate hypotheses that we later investigated using more rigorous methods; it was not a formal hypothesis test, in and of itself; lines 127-129, lines 690-693.
- 4) We added support for our conclusions regarding taxa other than molluscs and arthropods (e.g., vertebrates) by applying an additional analytical method, presented in a new Figure S5; lines 160-171, and SI lines 46-50 and Fig. S5.
- 5) We better explain the pseudo-methyltransferases, and how they support the evolutionary story, in the text and in a modestly revised Figure 4; lines 207-211 and replaced Fig. 4.
- 6) We more clearly present the evidence supporting an HGT origin for arthropod PKS genes; lines 283-289.
- 7) We reanalyzed the arthropod KS domains in several ways, generating a new Fig. S3, and modified the text accordingly; the revised results provided even stronger support for conclusions about distinct AFPK clades associated with insects, beetles, and spiders; lines 311-320, SI lines 31-35 and Fig. S3.
- 8) After an additional literature review, we synthesized data from molecular genetic studies showing insect genes in AFPK clades are associated with unusual lipid production. This new information adds to our knowledge of AFPK function, aside from the biochemically characterized members EcPKS1 and 2; lines 25-26, 52-54, and 327-335.
- 9) We created a new global KS tree (new Fig. 7) that strongly supports our conclusions as initially described, incorporating the major arthropod as well as mollusc AFPK sequences as suggested by the reviewers. This new tree is also more fully explained in a new section of text describing the evolution of enzyme function for these gene families; lines 353-361, 726-728, Fig. 7, SI lines 37-42 and Fig. S4.
- 10) Because of reviewer concerns about what we know and don't know here, we

moved any limitations to a “limitations” section, rewriting the text to be clearer; lines 399-413.

11) We modified the SI to include all the accession numbers used in this study (included within SI datasets).

12) We made all other requested changes throughout the manuscript.

Reviewer #1 (Remarks to the Author):

This manuscript seeks to identify fatty acid synthase-like polyketide synthases (AFPK) and understand their relationship to Type I Fatty acid synthase and Type I polyketide synthases in animals. The structural similarity between animal fatty acid synthase and Type I polyketide synthases is striking, and it is a pleasure to see an analysis of their evolutionary relationship.

Overall, the data and approach in this manuscript are compelling, but the description of the background and significance of the conclusions is too limited.

→ Thank you for your comments. They were extremely helpful in creating a much more complete overview of lipid metabolism in the animals. We have modified these sections as requested.

In particular, the title suggests that the paper will focus on the PKS origin of animal fatty acid synthase. This idea is supported by Figure 7, which includes the mollusk AFPK. However, the tree in Fig. 7 does not include the arthropod AFPK or type II fatty acid synthase. Rooting the tree based on PKS from Archaea and bacteria seems sensible, but it assumes, rather than tests, the idea that animal FAS is derived from an ancestral PKS. If there is data in the literature supporting the PKS origin of animal FAS, then it should be described more fully in the introduction. If not, the authors should add additional analysis to reject the null hypothesis that enzymes involved in specialized metabolism (PKS) evolved from enzymes in central metabolism. Alternatively, the authors could change their title to reflect the main focus of the manuscript, the identification of the AFPKs.

→ This is a great comment. We generated a revised tree as suggested, including both arthropod AFPKs and type II FAS. We replace the old Figure 7 with a new KS phylogeny that supports the hypothesis that fungal PKSs and animal FASs share a common ancestor, and also addresses some of the additional comments below. In addition, we modified the title to be clearer.

In addition, the manuscript either lacks sufficient background discussion supporting the identification of AFPKs as a separate class of polyketide synthases, or they underemphasize the significance of identifying this group of enzymes. It is unclear from the introduction whether they are merely expanding the identification of a previously known class of PKS enzymes or demonstrating that AFPKs, of which only one or two examples are known, are a widespread, uncharacterized subclass of PKS enzymes. The latter appears to be the case, given that the only AFPK discussed in the introduction was discovered by the authors' lab.

→ To address this issue, we have rewritten the abstract and introduction to clearly distinguish previous knowledge from new findings; lines 12-22, 39-70. Further, we expanded the conclusion section to emphasize our results in terms of current knowledge; lines 418-424.

Furthermore, it is unclear whether the AFPKs should be considered together as a class separate from the Type I PKSs.

→ The new tree produced in response to review definitively shows that the animal AFPKs and FASs comprise a clade that is derived with respect to the more distantly related PKSs. Further, the animal FASs are a monophyletic branch within the AFPK/FAS clade.

→ In addition, in response to this criticism, we were inspired to look more deeply and critically at the arthropod AFPKs identified in our study using informatics methods, a somewhat daunting task given the thousands of enzymes we identified. We found that some genes in ar-clades 2 and 3 were previously associated with insect specialized metabolism in cuticular hydrocarbon and pheromone biosynthesis, although biochemical studies are still lacking. These enzymes are called “FAS” in the literature, but are clearly AFPKs in our analyses. This literature is now described in the revision and highlights the need for a reexamination of past insect FAS studies in light of our findings; lines 25-26, 52-54, and 327-335.

In the phylogenies and dot plots (like fig. 6A), the AFPKs form multiple clades rather than clustering together. There was also little discussion about the relationship between the arthropod and mollusk AFPKs. Aside from the unrooted trees in Fig. 6, little was done to determine if these two groups of AFPKs have a common origin or evolved independently in different animal lineages.

→ This question was answered in a number of new analyses presented in the revision and described below. Most importantly, the revised Fig. 7 clearly shows that the AFPK family diversified early in protostome evolution and then specialized clades evolved in different hyperdiverse animal lineages.

Addressing the weaknesses listed above and those in the list below will provide clarity to this otherwise very interesting work.

Below are specific questions and comments that should be addressed:

Major comments:

p. 2, abstract, line 6: I don't understand the sentence “Vertebrates lack AFPKs, but excepting the placental mammals contain PKSs.” Overall, the abstract also doesn't clearly differentiate

between what was previously known and what this manuscript aims to demonstrate, as mentioned regarding the introduction above.

→ **This has been addressed in the revised abstract.**

p. 2, introduction paragraph 2: This paragraph suggests that AFPKs are a known class of enzymes that are distinct from Type I PKS, but the reference appears to only identify one or two from a single species. It also suggests that the AFPKs are a single group of enzymes that are distinct from the PKS enzymes, but their domain structure and data in Fig. 2 and 7 raise questions about whether they should be considered a subtype of the Type I PKS. In addition, the introduction suggests that the AFPKs are a single class of enzymes, but the data don't clearly show whether the mollusk and arthropod AFPKs have a single origin, since the arthropod clades are not in Fig. 7.

→ **In response to review, we better summarize why AFPK/FAS branch of metabolism is unique and should have its own designation. The revised Fig. 7 shows that neither mollusc nor arthropod lineages are monophyletic within the AFPK group, indicating an early diversification of AFPKs in protostomes and subsequent specialization.**

p. 4 and figure 2: The description of Figure 2 implies that the AFPKs are a known class of enzymes that are clearly defined by the data shown in Figure 2, suggesting that the purpose of the figure is to test an approach to annotate AFPKs, rather than experiment to see if AFPKs are likely to be abundant. When describing Figure 2, wouldn't it be more accurate to say that, in contrast to vertebrates, a significant number of KS domains in arthropods and mollusks score poorly against the 3 HMMs, suggesting that AFPKs are widespread in these species? Because you acknowledge that this method has lower power to discriminate among clades than phylogenetics, shouldn't the results be described as putative or probable AFPKs?

→ **Good point, change has been made in the description in the figure legend and in the text. We did indeed use this as a rapid method of data exploration to collect putative AFPKs, FASs, and PKSs, and to generate a hypothesis about which enzymes constitute the AFPKs. Another significant objective of this analytical approach is to facilitate the selection of KS sequences that are most likely to be AFPKs, thereby reducing the total number of KS sequences for downstream phylogenetic analyses. We then tested those hypotheses by extensive analyses described subsequently in the manuscript; lines 127-129, lines 690-693.**

Figure 2 clearly shows that a lot of KS domains have intermediate scores between FAS and PKS HMMs, but is the data really sufficient to define cutoffs to declare which ones are AFPKs, as opposed to more divergent FAS or PKS? In arthropods, some domains within the AFPK group have better PKS HMM scores than FAS scores, and some domains labeled as PKS in the mollusk group also have low scores for the PKS HMM.

The rationale for defining the score boundaries between AFPKs and FAS or PKS should be stated, particularly for arthropods.

→ As noted above, we did not believe that the data shown in Figure 2 were sufficient to delineate enzyme families unambiguously; this initial screen provided the hypothesis that we rigorously tested in the remainder of the study. This is now better described in the Figure legend and text; lines 127-129, lines 690-693.

Also, could the lower HMM scores of putative AFPKs in mollusks and arthropods be due to technical rather than biological reasons? For example, using a larger number or a less diverse set of vertebrates in the HMM training sets could weight the HMMs to more accurately identify vertebrate FAS and PKS and result in intermediate scores in the other organisms. Lower scores against the HMMs could also arise from truncated KS domains arising from incomplete transcriptome data. How was this taken into account? Could the observation that domains with intermediate scores are absent from the vertebrate data be due to the higher quality of that data?

→ We did not use truncated KS domains in our analysis, but only full-length sequences. The methods by which we obtained high-quality sequences are now better described in the first section of results. This section was rewritten to clarify the pipeline used in the analyses; lines 82-97.

→ To investigate the vertebrate HMM question more fully, we examined how different KS training sequences from less diverse species affected the medium score of AFPKs. To do this, we downloaded all protein sequences annotated as FAS or polyketide synthase from GenBank. Afterward, we extracted the KS domains, removed redundancy, and subjected them to analysis using an ML tree. Consistently, we only detected FAS and animal polyketide synthases (PKS) in our analysis. Subsequently, we constructed new HMMs for FAS and animal PKS using the newly detected sequences exclusively from vertebrates sourced from GenBank. These fresh HMMs were then utilized to generate HMM score dot plots for mollusks and vertebrate KSs. Remarkably, the resulting HMM score plots are very similar to the ones in Figure 2, including the relative placement of EcPKS1 and EcPKS2, reinforcing the robustness of the analytical method; lines 150-161, and SI lines 46-50 and Fig. S5.

p. 4, paragraph 3, regarding the statement that no AFPKs were identified in vertebrates: It would be better to say that no KS domains had intermediate bit scores to FAS I, PKS or FAS II, supporting the absence of AFPKs in these species. (Alternatively, the HMMs were weighted toward vertebrate proteins, so the scores are higher)

→ Based on the original manuscript, this would be a valid interpretation, but our additional analyses support the interpretation that there are no AFPKs (as defined in our paper) in available vertebrate datasets. In response to reviewer comments, we looked at this in many ways, which are now summarized in the section, “Widespread distribution of diverse, KS-containing type I enzymes in animals”; lines 160-171.

p. 7, paragraph 4, in the sentence “This spider clade was closely related to mollusc AFPKs, reflecting the ancient origin of AFPKs prior to the divergence of major bilaterian lineages”: I am not convinced that the relationships among AFPK clades are clear among these lineages in figure 6b. It does not appear that the spider clade is monophyletic with any particular mollusc AFPK clade or that the evolutionary distance between them is shorter than between other clades (the branches leading to both groups are long).

→This has been clarified in the updated phylogeny (revised Figure 7) including extensive additional sequence data as suggested.

p. 7, paragraph 4 and figure 6C, around the sentence “Although there are some overlapping regions between clades, the FAS KSs mainly localized in the right top of the plot.”: The discussion about fig. 6C is problematic. Ar-clade 3 obviously is more separated (divergent) from FAS because the y-axis HMM was constructed from ar-clade 3 sequences. What is most noticeable about figure 6C is that there is poor separation between FAS and ar-clades 1, 2 and 4, indicating that they are not more similar to ar-clade-3 than FAS. This fits with the phylogeny in fig 6b, where the clades of AFPKs do not cluster together. Aside from clade 3, the other clades and FAS are nearly or completely on the diagonal, indicating that all of them (including FAS) score nearly equally well with both HMMs. Other than showing that the subdivision of the arthropod FAS and AFPKs are ill-defined (also as shown in fig 2b), it is unclear how to interpret this figure or what conclusions should be drawn from it. In addition, the authors should comment on the grey dots, most of which score poorly against either HMM. Are these more divergent groups of FAS or AFPKs? Are they PKS?

→ Thanks, we agree that the dot plot is not a good way to determine whether the four AFPK clades represented all groups of AFPKs in available arthropod transcriptomes. Beside the phylogenetic tree2 in Figure 6, in response to review we also randomly divided 6,542 nonredundant Arthropoda KSs into 11 groups; each group was analyzed by ML tree using ar-clades1-4 as reference. The result strongly supported our conclusion. See the description in the section “AFPKs from arthropods”; lines 311-320, SI lines 31-35 and Fig. S3.

Minor comments:

Supplemental data files: It would be helpful if the supplemental data sets were given informative names. Accession numbers of the sequences in the data sets should be included.

→ Titles were provided, and sequence names were added to the supplemental data sets with relevant accession numbers.

Fig. 2: mark where EcPKS 1 and 2 are in fig 2a.

→We added the hmm scores for EcPKS1 and EcPKS2 to the legend; the score indicates the position in the figure; line 692.

p. 4 paragraph 3: Fig.2c is labeled as vertebrate KS, not chordate. If that is correct, then it is unclear why non-vertebrate chordates (tunicates) are discussed in this paragraph.

→ **We investigated all GenBank tunicate SRAs and found only FAS genes. The low complexity of KS-containing genes did not necessitate the HMM approach. This has been clarified in the paragraph; lines 142-148.**

p. 4 paragraph 3, second to last sentence concerning eukaryotic parasites. The data supporting the statement that PKS in eutherian mammal data sets are actually from parasites needs to be shown and described, particularly since the accession numbers that are shown list the source as sperm whale.

→ **In response to review, we reinvestigated these sequences. The proteins in question were recently removed from public databases as a result of standard genome annotation processing. This is likely because those proteins were automatically identified as resulting from contaminating parasites. With this change to the data, blastp searching for those proteins now exclusively hits proteins from animal parasites (apicomplexans). This validates our previous analysis described in the paper, but since the sequences have since been removed from animal datasets, we have removed this section from our paper.**

p. 5 paragraph 2: It is not clear what the “remaining sequences” refers to. Are you referring to sequences whose FAS HMM scores were <400 and >600? Or are you referring to sequences with scores between 400-600 that were not used to build the phylogeny? What were the upper and lower bounds?

→ **Clarified that this refers to all of the KSs found in the molluscs.**

p. 5 last paragraph through the beginning of page 6: The discussion of methyltransferase domains is confusing. An MT domain is only noted in clade 5. Are you saying that partial/degraded MT domains are detectable in other clades? If so, then these pseudogene exons should be labeled in fig 4b in order to evaluate and support the last few sentences of this paragraph.

→ **The “pseudo-methyltransferase” is structurally important in animal FAS and in PKSs, such as in the fungal PKSs. It is often not labeled in domain diagrams because it has no catalytic function and often has low similarity to functional methyltransferases. We have clarified this discussion by naming it the “pseudo-MT” in this section of the paper and describing it briefly with a reference; lines 207-211.**

p. 7, paragraph 2, in the sentence about horizontal gene transfer: What is the justification for claiming that some putative AFPKs in arthropods are derived from horizontal gene transfer?

→ These PKS genes included some that were previously identified by another research group as horizontally transferred PKS genes in the *Folsomia candida* genome. Therefore, we hypothesized that in this group, unusual PKS genes are all derived from horizontal gene transfer given prior findings to that effect. This has been clarified in the revised text; lines 283-289.

p. 7, paragraph 3, in the sentence “The dot plot of HMM bit scores (Fig. 6A) showed a very similar trend to that for the whole arthropod KSs, but with a much slower descent.”: Rather than “slower descent”, the wording in the figure legend is more clear - there is a sharper breakpoint between FAS and likely AFPK.

→ Now described in same way in the text, as requested; line 294.

p. 15 figure legend 1: change “are encoded on individual domains” to “are encoded on individual proteins”

→ Done; line 674.

p. 15 figure 1B: Some of the chemical structures don't convey the chemical reaction being described. In particular, the substrate of KS is shown as malony-ACP, (3 carbons), and its products are shown as CO₂ + a 4-carbon beta-ketoacyl-ACP (5 carbons), without explaining where the other carbons come from. Also, the product of ER is implied to be a polyunsaturated fatty-acyl-ACP, instead of a fully reduced fatty-acyl-ACP

→ We have redrawn the chemical figures as requested. It is harder to convey the iterative nature and the variation in the new diagram, but we have done our best; Fig. 1, line 670.

p. 20, figure 6 legend. In the sentence “Scatter plot of the arthropod KS bit scores to arthropod AFPK ar-clade 3 KS HMM and FAS HMM, showing that selected KSs are representative of the diversity of KSs found throughout the arthropods.” Given the large number of low-scoring sequences in gray, which were not included in the phylogeny, the selected KSs used in the phylogeny don't seem that representative of the arthropod KS domains.

→ The gray dots within the colored clades represented the same enzymes in those families, while the gray dots to the lower left represented PKSs and type II KSs. You are right, this is confusing. We have replaced it with a figure that does not have this weakness; Fig. 6, line 717.

p. 21, figure 7: the left section of the tree is labeled AFLP instead of AFPK.

→ Fixed in the new figure. We initially named these AFLPs, but upon realizing that was an older technique in the molecular literature, we switched to AFPK to avoid any confusion. Thank you for catching the inconsistency, we have fixed it, and sorry for the

confusion.

Reviewer #2 (Remarks to the Author):

This manuscript provides new information on the distribution of polyketide synthase and fatty acid synthase genes across broad scope of life forms. In particular, it focuses on animal fatty acid-like polyketide synthases or AFPKs. These biosynthetic systems generate a diverse spectrum of secondary metabolites thought to be important for self-protection, signaling and other communication functions.

The work will be of high significance to the field of natural product sciences, data science/computational analysis of metabolic systems, biochemists, and chemists. The work is highly original and provocative. It may very well provide a road more for many future studies to investigate pathway gene expression, enzymology of the transformation, and further predictive tools to understand the chemical diversity programmed by these fascinating PKSs.

There is some speculation relating to evolution of these systems, but the work provides important hypotheses and conceptual starting points for future studies that demand further detailed studies.

No fatal flaws were identified. While, the noted correlation between loss of mollusk shell and evolution of protective secondary metabolites seems logical, tying it directly to specific types of gene loss beyond the shell would provide a deeper, more sophisticated analysis. Overall the methodology is sound and meets the standards in the field. Sufficient detail is provided in the methods section.

The Conclusion paragraph is excessively brief. It requires more scholarly discussion about the major findings and limitations of this study.

→ Great suggestion. In retrospect, this was a major limitation. We have clarified the limitations of the study in a new section, and highlighted the novelty of our main findings; lines 399-429.

→Other changes. We searched the current NCBI database and updated the current KSs hits number in the main text, we also included the KSs that were excluded during redundant sequences removal.

REVIEWERS' COMMENTS

Reviewer #1 (Remarks to the Author):

The authors made an excellent effort to respond to the initial review. The additional analyses required a significant amount of time, and they increase the robustness of the analysis. The revisions of the text provide additional context and clarity. The authors convincingly demonstrate that the AFPKs constitute a new class of FAS and PKS-like natural product synthesis enzymes, which are ripe for biological and biochemical analysis.